# Transition, Adjustment, and Healthcare Avoidance: African Immigrant Women’s Experiences and Perceptions of Navigating Primary Healthcare in the USA

**DOI:** 10.3390/healthcare12151504

**Published:** 2024-07-29

**Authors:** Gashaye M. Tefera, Mansoo Yu, Erin L. Robinson, Virginia Ramseyer Winter, Tina Bloom

**Affiliations:** 1College of Social Work, Florida State University, Tallahassee, FL 32306, USA; 2School of Social Work, University of Missouri, Columbia, MO 65201, USA; yuma@missouri.edu (M.Y.); robinsonel@missouri.edu (E.L.R.); 3Department of Public Health, University of Missouri, Columbia, MO 65201, USA; 4School of Social Work, University of Minnesota, Minneapolis, MN 55108, USA; ramseyerwinterv@missouri.edu; 5School of Nursing, Notre Dame of Maryland University, Baltimore, MD 21210, USA; tbloom@ndm.edu

**Keywords:** access to healthcare, transition and adjustment, immigration, African immigrants, Ethiopian immigrant women

## Abstract

This study explores the transition and adjustment of African immigrant women, particularly Ethiopian immigrant women (EIW), as they navigate the U.S. healthcare system and their ability to access and utilize healthcare services. A qualitative cross-sectional design with a mix of purposive and snowball sampling techniques was utilized to recruit EIW (*N* = 21, ≥18 years) who arrived in the U.S. within the last five years. One-on-one in-depth interviews were conducted to collect data. The interviews were audio-recorded, transcribed verbatim, and analyzed thematically using Nvivo12 software. The thematic analysis revealed three major themes: (1) settling into new life in the U.S. delays EIWs’ ability to access primary healthcare; (2) adjusting to the U.S. healthcare system: confusions and mixed perceptions; and (3) avoidance of care: EIW’s reasons for PHC visits changed in the U.S. Participants avoided healthcare, except for life-threatening conditions, general check-ups, and maternal healthcare services. Transitional support for legal, residential, employment, and health information could help tackle the challenges of accessing primary healthcare for EIW. Future research should analyze access to healthcare in relation to the everyday struggles of immigrant women, as well as legal and complex structural issues beyond acculturative issues.

## 1. Introduction

Navigating primary healthcare (PHC) in a new country is a major challenge experienced by immigrants [1]. Oftentimes, their ability to navigate and access PHC is constrained by barriers such as language, financial limitations, lack of adequate knowledge, insurance costs, and discriminatory sociopolitical environments [2,3]. For example, 25% of lawfully present immigrants and 46% of undocumented immigrants are uninsured, compared to 8% of U.S. citizens [4]. These barriers may contribute to immigrant health disparities, including higher rates of cardiovascular diseases, such as hypertension, diabetes [5], and infectious diseases [6].

Recently arrived immigrants (i.e., <ten years) are particularly likely to lack access to PHC [7]. Research documents how the adjustment to living in a new country significantly affects recently arrived immigrants’ ability to seek healthcare services, acting as another burden in their ability to access care [8]. Most studies center around linguistic and cultural adjustments as barriers and use the umbrella term ‘acculturation’ (i.e., assimilation into the dominant culture) to describe transitional challenges [9,10,11]. However, this emphasis on acculturation often ignores the impact of non-cultural forces in limiting immigrants’ ability to access PHC services. It also puts the burden on individuals and their adaptive abilities instead of the complex structural and systemic barriers that put pressure on access to healthcare [12].

Additionally, many studies focus on challenges that immigrants experience after they have established contact with providers in physical healthcare settings [13,14,15], e.g., communication barriers, the cultural insensitivity of professionals, limited health literacy, and expensive insurance/costs [16]. Although these challenges are important, there is also a strong need to investigate barriers that immigrants experience before encountering the PHC system, including legal and bureaucratic aspects of immigrants’ lives and everyday barriers that have proxy effects on healthcare access [17]. Similar to their transition from their country of origin to a new country, they must adjust to a new healthcare system that is different from the system in their home country. Differences in healthcare systems between countries of origin and immigrant host countries are a very real barrier for many immigrants, yet its impact is understudied [2].

Finally, although the U.S. immigrant population is substantially increasing and highly diverse, most studies have focused on Asian and Hispanic immigrants [1]. African immigrants are often included in the singular “Black” category in the immigrant health literature; thus, their healthcare experiences and needs are often misrepresented or completely ignored [18]. In addition, African immigrant women have the least access to healthcare and the highest diagnosis of STDs and HIV/AIDS [19,20]. Thus, there is a strong need to explore African immigrants’ unique healthcare needs and experiences, particularly women, to inform practice and policy [21].

Therefore, the current study aimed to address these gaps by exploring the transition and adjustment of African immigrant women, particularly Ethiopian immigrant women (EIW), as they navigate the U.S. healthcare system and their ability to access and utilize healthcare services. Furthermore, this study extends beyond the ‘acculturation’ narrative to focus on additional structural challenges in their transition and adjustment into the U.S. and the healthcare system.

### The Context of PHC in Ethiopia

In Ethiopia, PHC is free in public health centers, clinics, and hospitals. There is no health insurance system, and most people are unfamiliar with the concept. Instead, patients are expected to pay a few Ethiopian Birr for a patient identification card (i.e., card number) and then can be examined/treated at no cost in public health hospitals/centers/clinics. Prescribed medications can be purchased from the public (at a lower price) or private pharmacies. In some cases in which advanced testing is required, healthcare costs in Ethiopia can increase and become quite costly for many people. Although most of the population uses public PHC services, there is a growing trend of accessing private healthcare in the country’s major cities. This is mainly due to the overburdened public healthcare centers and the relatively fast and better health services in private healthcare centers. All payments are made in cash and upfront in both public and private health centers. There is no health insurance system except for a few employers (mostly international organizations) with medical allowances for their employees [22,23]. Hence, there are many differences between the Ethiopian and U.S. healthcare systems, and EIW are well-positioned to describe adjustment challenges in light of those differences.

## 2. Methods

### 2.1. Research Design

We used a qualitative, cross-sectional design with a phenomenological approach to explore EIW’s experiences, perceptions, and challenges in adjusting to the U.S. and their ability to access healthcare services. The use of a qualitative approach was appropriate due to the following: (1) the unavailability of adequate information on the healthcare experiences of EIW; (2) the need for an in-depth examination of the complex realities involved in EIW’s lived experiences [24].

### 2.2. Sampling and Recruitment

Due to the hard-to-reach nature of this population, a mix of purposive and snowball sampling was used to recruit participants who (1) identified as an Ethiopian immigrant; (2) identified as female; (3) were ≥18 years old; (4) arrived in the U.S. within the last five years; and (5) spoke Amharic and/or English. According to Creswell and Clark (2018), a sample of five to twenty-five participants is adequate for qualitative phenomenological studies. Two participants were selected to pilot the interview questions and test their appropriateness and clarity. After the pilot, 21 participants were recruited through community and service organizations that served the immigrant population. Recruitment flyers and word of mouth helped to reach potential participants [25].

### 2.3. Procedures

After the study proposal was approved by the Institutional Review Board (IRB) of the University of Missouri (IRB Review Number 378001), data were collected through one-on-one, in-depth interviews (*N* = 21) that lasted between 45 and 90 min. A pilot-tested interview guide with open-ended questions, and a structured demographic questionnaire was used to collect data. The pilot-testing of the interview guide helped us to revise some of the questions, avoid redundancy, add probing questions, and restructure the order of the questions. Interviews took place in person (*n* = 5) and virtually (via phone and Zoom, *n* = 16) based on participants’ preferences and locations. All the in-person interviews took place privately in the participants’ residences. At the start of each interview, the lead investigator reviewed the study information with each participant and gained verbal consent. All interviews were audio-recorded and stored securely on a password-protected computer. Interviews were transcribed verbatim and identifiable information was removed from transcripts to ensure confidentiality. Pseudonyms were used in place of participant names to further protect participants’ identity.

### 2.4. Analysis

We used an inductive thematic analysis process to allow dominant themes and categories to emerge from the raw data [26]. A five-step bottom-up approach was followed: (1) data were organized, transcribed, and cleaned; (2) deidentified transcripts were read repeatedly, and memos were kept; (3) potential themes and patterns were built, and a codebook was used to code each transcript in Nvivo12 software; (4) interpretations were assessed and developed; and (5) a structural and textural description of that which EIW experienced in accessing healthcare was provided [24]. Interviews continued until saturation was reached. Prolonged and persistent engagement with data, interview notes, and memos, as well as thick and thorough descriptions and direct quotes from participants, were used to increase rigor and trustworthiness [27].

## 3. Findings

### Demographic Information

Participants (*N* = 21) were current residents of six U.S. states (MO, MD, VA, KS, MI, and IN) and Washington DC, and ranged in age from 24 to 53 years (*M* = 36.6). The majority of participants had asylee or asylum-seeking status (*n* = 8), were married (*n* = 12), and had used private insurance (*n* = 15). See Table 1 for detailed demographic information.

We found that EIW in this sample endured two major adjustments, which included settling into life in the U.S. and adjusting to a new healthcare system. Both caused delays in access to PHC and avoidance of healthcare. The thematic analysis (Figure 1) revealed three major themes: (1) settling into the new life in the U.S., causing delays in EIWs’ ability to access PHC; (2) adjusting to the U.S. healthcare system with confusions and mixed perceptions; and (3) avoidance of care, where EIW’s reasons for PHC visits changed in the U.S. In addition, four distinct sub-themes emerged under the second major theme (see Figure 1).

## 4. Theme 1: Settling into the New Life in the U.S. Delays EIWs’ Ability to Access PHC

Participants stated that the overwhelming experience of settling into the U.S. would not allow them to prioritize their healthcare needs. The first couple of years were filled with stressful events: familiarizing themselves with the new environment, finding a place to live and permanent residential addresses, adjusting their legal status, processing identity documents, obtaining work permits, finding jobs, and understanding how things work in the U.S. Hence, most EIWs did not attend to their PHC needs in these years of adjustment. For example, Haben (all names are pseudonyms) shared the following:

It was tough; I didn’t know anyone, and it took me almost two years to adapt and connect with people and learn about how to have and use insurance and healthcare. This is in the DC-Maryland-Virginia (DMV) area, where many Ethiopians live. I can’t imagine how it will be difficult for those living in other areas with no Ethiopian community.(Haben, age 36)

Settling into life in the U.S. was not a straightforward path. Many participants continually changed their legal status and residential addresses, disrupting their ability to become employed and stay connected to PHC services. For example, Saba changed her legal status four times and lived in three different states within her first two years in the U.S. She said the following:

I was on a student visa in Iowa. Then, I moved to California on an OPT visa and then changed to an H1B and moved out of California on a different visa. The project ended, and I was suddenly at risk. I had to change my status immediately and had to move out of California, and I stayed without insurance.(Saba, 35 years old)

Most participants agreed that the first two years were the most challenging years in terms of adjustment and their ability to pursue PHC services. Maya, 38 years old, said, “The first year and a half, I lived without any insurance, health service, or check-ups at all. I had so many health issues, but I had to wait until my documents were processed and I got employed”. Leaving their families and support systems behind and navigating an individualistic lifestyle in the U.S. affected EIW’s ability to seek and utilize PHC services. In particular, participants who gave birth in the U.S. were affected by the limited social support system. Salem said the following:

I was pregnant with no one around. My husband is still in Ethiopia. It was the toughest time of my life to be a single mother in a new country, with no job and healthcare for a while until I processed documents and learned the language.(Salem, 37 years old)

The hassle of adjusting to the weather, homesickness, poor working conditions (including standing for long hours), communication challenges, and the lonely lifestyle in the U.S. hindered EIW’s ability to prioritize their well-being or healthcare needs. The emotional burden of adjustment and limited knowledge constrained EIW’s capacity to navigate the PHC system effectively. Most importantly, EIW’s experience depicted the lack of organized institutional or systemic support in helping immigrants adjust to their new lives, which could cause part of the problem of accessing PHC.

## 5. Theme 2: Adjusting to U.S. Healthcare System: Confusions and Mixed Perceptions

Participants shared their struggles in adjusting to the U.S. healthcare system and perceptions of PHC services compared to their experiences of PHC in Ethiopia. Four subthemes emerged under this category:

### 5.1. Subtheme 1: The Nature and Procedure of Primary Healthcare System

Participants were expected to unlearn Ethiopia’s healthcare procedures and struggled to familiarize themselves with the U.S. system. Coming from a developing nation with relatively universal healthcare, EIW expressed that it was difficult to comprehend why basic primary healthcare was not universally available in a wealthy country like the U.S. Most participants expressed their shock when they realized they could not access services without paid insurance and healthcare is very expensive:

I was shocked to learn about healthcare expenses. In Ethiopia, if I don’t have money, I would go to a public health center to get my treatment and contraceptive pills for free. I lived in the anxiety of getting pregnant here because I couldn’t afford it as a student. It is insane given the wealth and advancement of the U.S. that you have to struggle to meet basic PHC needs.(Hager, 30 years old)

Familiarizing themselves with the healthcare and insurance system was an enormous challenge. As Gelila, 53 years old, put it, “It is a whole new world”. Participants stated that it took them some time to realize that the U.S. healthcare system is vastly different than their home country. Most of the participants admitted they still do not completely understand the complexity of the insurance and PHC system. Hager, 30 years old, said, “I still rely on my husband. It is a lot to figure out”.

For participants who could formerly visit any nearby Ethiopian health center when they felt sick, meeting the expectations of insurance companies, identifying the in-network providers, understanding copayments and deductibles, covered and uncovered conditions, obtaining specialist referrals, and making appointments proved overwhelmingly difficult. “There are so many requirements and expectations that overburden you as a patient. In Ethiopia, if I get sick, I will just get up and go to the nearest hospital. Not anymore”, said Hager, 30 years old. The standard and procedures of care in the U.S. are a huge learning curve for EIW. Most participants stated that regular check-ups, screening, and the highly specialized practice that involves multiple referrals are new to them.

### 5.2. Subtheme 2: U.S. Healthcare Is Higher Quality, Yet Inaccessible

For participants, PHC in the U.S. was characterized by its two opposite features: better quality but inaccessible. Although participants appreciated the universal nature of Ethiopian PHC and its relative affordability, they also remembered overburdened healthcare centers, with long lines, waiting hours, limited human resources, inadequate medical supply and equipment, and the unavailability of treatment beds and rooms for admission. Gelila, 53 years old, said, “In Ethiopia, if you need to be admitted, it would be very hard to find a spot as the number of patients outnumbers the available places and physicians”. A midwife-nurse herself, Samiya, 34 years old, offered this contrast: “My second baby was born here (U.S.) before the due date and had to stay in ICU for weeks. I didn’t have to worry about infections, which are the main cause of health complications back in Ethiopia”.

However, the inaccessibility of the U.S. care system causes frustration. Ametsash, 53 years old, used the expressed on an Amharic poetic proverb, “ላም አለኝ በሰማይ ወተትዋንም አላይ”, which translates to “I have a cow in the sky, I will never see (get) her milk”, which is a saying used to refer to lack of hope. Saba, 35 years old, added, “As to me, the PHC service is very good; the treatment and follow-up are good. However, the cost is unspeakable. You are left out if you do not have a well-paying job”. For most participants, the inaccessible nature of PHC in the U.S. is stressful. Maya, 38 years old, said, “Leaving my job with medical allowance and all the privileges back home and struggling here to process immigration and get affordable insurance for my children has been emotionally draining”. Most of the participants repeatedly stated that they believe that PHC in the U.S. is primarily driven by making profits instead of saving lives. Hence, they have mixed feelings, and most of them approach the PHC services with suspicion and extra care.

### 5.3. Subtheme 3: Patient Involvement

Compared to their experience in Ethiopia, participants stated that PHC professionals in the U.S. try to find every piece of information and engage with their patients in decision-making and the care process. For some of the participants, this was appreciated but also difficult. First, EIW came from a culture where they mostly relied on a physician to examine them, identify the problem, and tell them what to do. As Halima, 33 years old, said, “Sometimes I feel confused when they ask me too many questions, and they explain too much information. Sometimes, I don’t even understand what they are saying as it is too technical for me”. Second, most EIW have limited English language skills, which makes communication stressful. But, for a few with a better command of the English language and technical understanding, this experience is positive. Regarding prenatal care, Hager, 30 years old, said, “The doctor explained the three options I have, and I liked the way she engaged me in the decision-making process”. A few of the participants also noted that their U.S. PHC experiences taught them an important lesson about the need to advocate for themselves in unfair situations. Meba, 32 years old, said, “After they misdiagnosed me, I learned that I have to advocate for myself and express my thoughts and feelings. I no longer stay too humble and quiet as I used to be back in Ethiopia”.

### 5.4. Subtheme 4: Recognizing Disparities in Access

Beyond their adjustment challenges and perceptions of overall inaccessibility, participants also recognized disparities in accessing PHC in the U.S. They believed that although PHC in the U.S. is generally inaccessible, the problem is greater among immigrants and other minorities:

When you are an immigrant in the U.S., having the courage to go to the health center is very difficult, even to enter the system. When I first came, I did not have a job or proper documents, and I was not even eligible to get government-sponsored insurance programs. Because of that requirement, I had to pay a lot of money, even my brothers had to send me money from home (Ethiopia) to get treated.(Halima, 33 years old).

Participants also stated that PHC is not accessible for people of color, particularly black immigrants. During the interviews, participants expressed directly and indirectly that their PHC experience had been influenced by their immigration status and racial identity. Marda, 35 years old, said, “So, I would say my experience is pretty much shaped by my understanding of the race relation in this country and how the entire medical industrial complex is racist and unfair to some people, and how it took me a while to understand that”. The interviews also revealed how participants perceived the residential and economic segregation in the PHC services in the U.S.:

Living in low-income communities means the healthcare facilities are poorly resourced. I was in Maryland, and I went to a hospital. The hygiene was so poor and packed with people waiting for services. I went to another hospital in an affluent neighborhood to visit a friend and it felt like I was in a different country. It was of high quality in everything.(Meba, 32 years old)

Participants also recognized the difference between private and public health insurance. They stated that although public programs offer lower-cost options, they are under-resourced, and the waiting time to obtain appointments and treatments is longer. Sishat, 41 years old, who used the Washington DC public insurance, said, “You should be lucky to get an appointment when using public programs. I had to wait for long hours for treatment, which was very challenging given my work schedule”.

## 6. Theme 3. Avoidance of Care: EIW’s Reasons for PHC Visits Changed in the U.S.

The interviews revealed that EIW’s reasons for seeking PHC have changed since moving to the U.S. Many developed a new habit of undergoing annual regular check-ups and screening in the U.S. Haben, 36 years old, said, “Here you do not have to be sick to go to health centers. You can go for check-ups and examinations as a preventive measure. That is good”. Next to check-ups, maternity-related services were the most common reasons for participants visiting PHC in the U.S. Thirteen out of the twenty-one participants stated they used PHC mainly during their pregnancy. Ruth, 32 years old, said, “I would say my top reason is pregnancy and pre- and post-maternal follow-ups. This is not something I can avoid, but for other illnesses, I prefer not to go”.

However, most stated that they did not pursue PHC for issues like headaches, stomach problems, or non-chronic illnesses as they used to do back in Ethiopia. They tried to avoid PHC if the issue was not severe or life-threatening. For example, only four said they regularly used PHC because of illnesses, including diabetes, thyroid, and back pain. The others would seek care only if severely ill or if the pain would not let them work.

Participants shared a broad range of factors related to avoidance of the PHC, including difficulty navigating the healthcare system, lack of satisfactory results, legal status, illegibility issues, lack of insurance, and financial limitations. Ametsash, 52 years old, said, “I wish I could find a solution for the number of health problems I have. I don’t feel well but I also don’t want to go as it is too expensive for me, and not satisfying results”. Although participants shared the overwhelming and stressful experience of adjusting to the new environment and healthcare system, only one was described using mental health services:

I started seeing a mental health provider since I came here, which is new in part because I wouldn’t say I never needed it while in Ethiopia, but I would say this life puts you through so much, and you need to make sense of it, and you need to process it. Therapy offers you a space where you can talk about your experiences and make sense of what happened to you. As I said when you go through workplace racism and racism in other spaces including healthcare.(Marda, 35 years old)

## 7. Discussion

Despite the significant presence of Ethiopian immigrants in the U.S., as the second-largest African diaspora in the nation [28], this is the first known study to examine EIW’s challenges in accessing PHC in the U.S. Our findings show that the PHC experiences of the EIW in this study are shaped by two major forces: (1) their settlement into life in the U.S., and (2) the transition from a relatively universal healthcare system to a non-universal healthcare system. First, before EIW even began to make sense of the healthcare system or started seeking PHC, they needed to figure out how to settle in and understand the life in the U.S. they had just moved into. Their main adjustment issues or priorities were processing or changing legal status, securing housing (permanent address), and employment (and work authorization). These issues are interconnected and structural and demonstrate that EIW’s challenges cannot be fully captured by the acculturation narrative that often puts a burden on immigrants and their cultural adaptive ability. As confirmed through a meta-synthesis of 83 studies [11], most studies on perceptions and experiences of immigrants in accessing PHC and healthcare in general focus on barriers that occur after contact is established with the healthcare system [29,30,31] as understood through the lens of ‘acculturation’ [10]. Although our findings do not contradict the importance of acculturation in immigrants’ healthcare experiences [11,32], they suggest that understanding immigrants’ access to healthcare requires examining non-cultural aspects of immigrants’ lives.

These findings support a previous study [33], which reported that immigrants in their first couple years of arrival find it extremely challenging to access and utilize PHC services. In addition to the legal, employment, and housing challenges, the EIW in this sample had to overcome mental and emotional challenges, such as homesickness and loneliness, resulting from leaving their families and communal lifestyle in Ethiopia behind and the stress of adjusting to an individualistic lifestyle in the U.S. This underscores the importance of viewing access to healthcare in relation to immigrants’ everyday struggle to adjust to their new life, and the need for an integrated approach in understanding the complex realities experienced by EIW within and outside healthcare settings [34].

Most EIW in the sample arrived in the U.S. with little or no information about the healthcare system, except the assumption that they will have better care given the country’s profile as a developed and prosperous nation. EIW also reported the absence of immigrant health information centers with the goal of serving the immigrant community. Our participants’ expectations were met with a different reality, similar to previous studies that documented immigrants coming from countries with universal healthcare systems, for example, Russian-speaking immigrants from the former Soviet Union [35], who struggled to navigate healthcare and find timely services [29,36]. However, research that investigates the impacts of transitions from one healthcare system to another is still limited, and understanding healthcare system-level differences between immigrant’s past experiences and their current situation is key to addressing immigrant’s challenges. Our findings further support the fact that providing pre-arrival health system information for immigrants is an important area for intervention versus current practice, i.e., only requiring the test results of infectious diseases (e.g., T.B.) from visa applicants or potential immigrants [37].

We found that EIW’s transition from a relatively universal healthcare system in Ethiopia to the U.S. system further shaped their overall perceptions and experiences. On the one hand, they found PHC in the U.S. to be advanced in technology and quality relative to Ethiopia; on the other hand, they found PHC in the U.S. inaccessible, especially for immigrants—similar to research with Brazilian immigrant women who were satisfied with the quality of healthcare services but found multiple barriers impeded their ability to access and utilize them [38]. EIW in this sample were new to the nature and procedure of the healthcare system in the U.S. and struggled to understand the standard procedures of having health insurance, identifying providers in their network, the need to make appointments, the coverage of prescriptions, and navigating referral systems. The findings suggest the need for programs tailored to address health literacy and information gaps among immigrants. Collaborating with immigrant community organizations and faith-based institutions to provide health education for immigrants could help expand the access and utilization of PHC among EIW.

The struggle to adjust to the new life and healthcare system in the U.S. places EIW under pressure to be highly suspicious and delay and avoid healthcare. Our participants reported avoiding healthcare for illnesses, which is consistent with findings on other immigrant women groups, including Latina and Asian immigrants [13,39]. They clearly adjusted their perceived needs since moving to the U.S. and limited their PHC visits to only critical conditions to avoid the burdens of accessing care. Unlike the consistently reported low rates of check-ups and health screening among immigrants [40,41], the most reported type of service used by EIW is a general check-up followed by maternal care. However, as the findings suggested, this should not be mistaken as progress because participants considered general check-ups as relatively the ‘easiest’ way to ensure they have no life-threatening issues but do not seek care based on the outcome of these check-ups. The situation also underscores the opportunity for general check-ups and maternity care visits as potential points of supportive interventions and the need to identify the further healthcare needs of EIW.

## 8. Limitations

Convenience sampling, especially snowball sampling, might have caused under-representation or over-representation of certain groups within the study sample. Our participants represented diverse immigration categories and different states and yet still shared common themes; however, unrepresented legal status or geographic differences (including local or state-based healthcare policies) could further impact PHC access or experience and warrant future investigation.

## 9. Conclusions

As the first known work of research on EIW’s challenges in adjusting to their new life in the U.S. and the PHC system, this study has important implications. The study underlines that immigrant women’s ability to seek PHC is highly influenced by two-directional struggles to adjust to their new life and as well as to the U.S. healthcare system, an adjustment which proceeds beyond cultural or acculturational aspects and extends to issues of legal status, employment, residence, and other structural factors that hinder their eligibility and ability to access and utilize PHC. It is imperative that future research analyzes access to healthcare in relation to the everyday struggles of immigrant women and legal and structural issues and investigates factors before contact has been established with the PHC. This study also demonstrates how the transition between different types of healthcare systems adds burden and further complicates immigrant women’s healthcare navigation efforts and underscores their need for customized pre-arrival and post-arrival information and transitional resources regarding the U.S. healthcare system. Overall, providing transitional support that includes support on legal, residential, and employment issues, resource mapping, and establishing immigrant health information centers could help tackle the challenges of EIW in accessing healthcare and reduce healthcare avoidance. Future research should focus on investigating each immigration category separately and focus on specific states or geographies as local or state-based healthcare policies vary across the U.S. Large-scale research that focuses on the impact of adjustment, legal and structural factors is warranted to expand the knowledge base of immigrants’ challenges in accessing healthcare.

## Figures and Tables

**Figure 1 healthcare-12-01504-f001:**
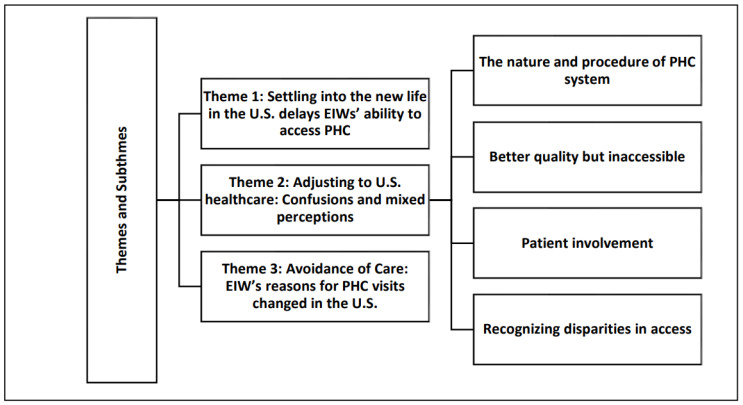
Themes and subthemes.

**Table 1 healthcare-12-01504-t001:** Demographic information.

Characteristics	Number/Percentage
Age	
18–25	1 (4.8%)
26–32	6 (28.6%
33–40	9 (42.9%)
41–47	1 (4.8%)
48–55	4 (19.05%)
Relationship status	
Married	12 (57.1%)
Single or never married	5 (23.8%)
Divorced or separated	3 (14.3%)
Widowed	1 (4.8%)
Lives with a partner	
Yes	11 (52.4%)
No	10 (47.6%)
Children	
Yes	13 (61.9%)
No	8 (38.09%)
Number of children	
1	5 (23.8%)
2	6 (28.6%)
3	1 (4.8%)
4	1 (4.8%)
Household size	
1	4 (19.05%)
2	3 (14.3%)
3	5 (23.8%)
4	5 (23.8%)
5	4 (19.05%)
Length of stay in the U.S.	
1–2 yrs	1 (4.8%)
3–4 yrs	9 (42.9%)
5 yrs	11 (52.4%)
Immigration status	
U.S. citizen	4 (19.05%)
Permanent resident (green card holder)	3 (14.3%)
Asylee	5 (23.8%)
Refugee	5 (23.8%)
Asylum seeker	3 (14.3%)
Pending (transition from J1 to O1, where J1 visa is an exchange visitor visa and O1 is a via category for individuals with extraordinary abilities or acheivement)	1 (4.8%)
Education	
Elementary	1 (4.8%)
Highschool graduate or GED	5 (23.8%)
Some college	1 (4.8%)
Graduated college	8 (38.09%)
Postgraduate study	6 (28.6%)
English proficiency	
Full professional proficiency (advanced)	2 (9.5%)
Professional working proficiency (intermediate)	12 (57.1%)
Elementary proficiency	6 (28.6%)
No proficiency	1 (4.8%)
Other languages	
Amahric	21 (100%)
Oromiffaa	3 (14.3%)
Tigrigna	1 (4.8%)
Somali	1 (4.8%)
Dawrogna	1 (4.8%)
Hadiyigna	1 (4.8%)
Kembatigna	2 (9.5%)
Swahili	1 (4.8%)
Arabic	3 14.3%)
Turkish	1 (4.8%)
Employment status	
Employed fulltime	17 (81%)
Employed parttime	3 (14.3%)
Not employed, looking for work	1 (4.8%)
Not employed, not looking for work	1 (4.8%)
Annual household income	
Less than 19,999	1 (4.8%)
20,000–39,999	8 (38.09%)
40,000–59,999	3 (14.3%)
60,000–79,999	3 (14.3%)
80,000–99,999	2 (9.5%)
100,000–more	4 (19.04%)
Residence	
MO	9 (42.9%)
MD	3 (14.3%)
VA	3 (14.3%)
Washington DC	3 (14.3%)
KS	1 (4.8%)
MI	1 (4.8%)
IN	1 (4.8%)
Health insurance	
Yes	18 (85.7%)
Private	15 (71.4%)
Public	3 (14.3%)
No	3 (14.3%)
Frequency of PHC use per year (non-pregnancy-related)	
1–2	14 (66.7%)
3–4	5 (23.8%)
5–6	2 (9.5%)
Common reasons for PHC visit	
General regular check-ups	15 (71.4%)
Pregnancy (pre- and post-natal)	13 (61.9%)
Dental regular check-ups	5 (23.8%)
Backpain	3 (14.3%)
Diabetes	2 (9.5%)
Vision	2 (9.5%)
Thyroid	1 (4.8%)
Tonsilitis	1 (4.8%)
Mental health service	1 (4.8%)
Overall self-rated health	
Excellent	5 (23.8%)
Very good	8 (38.1%)
Good	6 (28.6%)
Fair	2 (9.5%)

## Data Availability

The data presented in this study are available on request from the corresponding author. The data are not publicly available due to ethical restrictions.

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
