# Peer review of "Transition, Adjustment, and Healthcare Avoidance: African Immigrant Women’s Experiences and Perceptions of Navigating Primary Healthcare in the USA"

_healthcare, 2024, doi:10.3390/healthcare12151504_

Round 1

Reviewer 1 Report

Comments and Suggestions for Authors

Dear Author, 

the work is impressive and the study is focused. there are following required changes in this manuscript

·         The sampling selection of this research work is not clear. The author should mention the type of research with minimum sample selection criteria with included and excluded factors.

·         This research work is all about collection and information only , the treatment and strategy part is missing in this work. Try to highlight in results and discussion part.

·         Line 115: A pilot tested interview …. Is not clear and relevant for this study kindly justify

·         Thematic analysis of qualitative work is good, but the formula and application need to mention in the methodology.

·         Line no 151: Theme 1: Settling into the New…. That needs to be rewritten as per the methodology. Try to avoid repetition of words and work.

·         In line 324, the discussion part “This study aimed to explore the challenges of adjusting” is not required to be mentioned.

Comments on the Quality of English Language

Extensive English correction is required

Author Response

Please see the attached response

Reviewer 2 Report

Comments and Suggestions for Authors

1. Line 14 abstract: please check again the use of the words "were" or "was".

2. Where is the "Introduction" section written in this manuscript?

3. line 29: Why use the term "new country"? Are there no other terms that are relevant to this study? and is the US a "new country" in this study?

4. Line 74: Please find an alternative use of the word "new country" in the sentence

5. Lines 77-81: There are some citations there, but we don't see any reference sources listed! Please add reference sources!

6. Methods: Does this study require ethical clearance before data collection is carried out? Please explain!

7. Methods: Have participants in this study agreed and signed the information concern before data collection was taken? please explain

8.  Methods: Has the questionnaire used in this study been tested for validity and reliability? please explain

9.  Conclusion: Please add recommendations for further studies related to this problem

10. References: Please update the latest references, many old references have been found that can be replaced with the latest studies

11. References: Please check the reference writing format, especially reference number 36

Comments on the Quality of English Language

Minor editing of the English language required

Author Response

Please see the attached response

Reviewer 3 Report

Comments and Suggestions for Authors

TThis is an interesting article, but I have a few improvements to propose:

  1. In the methodology, the authors did not mention the paradigm they are using.

  2. Table 1 (Demographic information) should be in the Appendix, not in the main text.

  3. The format of citations should be consistent throughout. Right now it is a mess, with sometimes new paragraphs and sometimes quotation marks.

  4. Delete Figure 7 (by the way, where are Figures 2, 3, 4, 5, and 6?), as it is unnecessary. You are doing thematic analysis, not content analysis, so you do not need numbers.

Author Response

Please see the attached response 
